# Transformation of CO_2_ with Glycerol to Glycerol Carbonate over ETS-10 Zeolite-Based Catalyst

**DOI:** 10.3390/molecules28052272

**Published:** 2023-02-28

**Authors:** Zhangxi Gao, Mei Xiang, Mingyang He, Weiyou Zhou, Jiayao Chen, Jiamin Lu, Zeying Wu, Yaqiong Su

**Affiliations:** 1Jiangsu Key Laboratory of Advanced Catalytic Materials and Technology, School of Petrochemical Engineering, Changzhou University, Changzhou 213164, China; 2Research Center of Secondary Resources and Environment, School of Chemical Engineering and Materials, Changzhou Institute of Technology, Changzhou 213032, China; 3Xi’an Key Laboratory of Sustainable Energy Materials Chemistry, State Key Laboratory of Electrical Insulation and Power Equipment, School of Chemistry, Xi’an Jiaotong University, Xi’an 710049, China

**Keywords:** CO_2_, glycerol, glycerol carbonate, ETS-10 zeolite, cobalt-based catalyst

## Abstract

Catalytic conversion of CO_2_ with the surplus glycerol (GL) produced from biodiesel manufacturing has attracted much academic and industrial attention, which proves the urgent requirement for developing high-performance catalysts to afford significant environmental benefits. Herein, titanosilicate ETS-10 zeolite-based catalysts with active metal species introduced by impregnation were employed for coupling CO_2_ with GL to efficiently synthesize glycerol carbonate (GC). The catalytic GL conversion at 170 °C miraculously reached 35.0% and a 12.7% yield of GC was obtained on Co/ETS-10 with CH_3_CN as a dehydrating agent. For comparison, Zn/ETS- Cu/ETS-10, Ni/ETS-10, Zr/ETS-10, Ce/ETS-10, and Fe/ETS-10 were also prepared, which showed inferior coordination between GL conversion and GC selectivity. Comprehensive analysis revealed that the presence of moderate basic sites for CO_2_ adsorption-activation played a crucial role in regulating catalytic activity. Moreover, the appropriate interaction between cobalt species and ETS-10 zeolite was also of great significance for improving the glycerol activation capacity. A plausible mechanism was proposed for the synthesis of GC from GL and CO_2_ in the presence of CH_3_CN solvent over Co/ETS-10 catalyst. Moreover, the recyclability of Co/ETS-10 was also measured and it proved to be recycled at least eight times with less than 3% decline in GL conversion and GC yield after a simple regeneration process through calcination at 450 °C for 5 h in air.

## 1. Introduction

The concentration of carbon dioxide in the atmosphere has increased significantly by about 40% since the beginning of the industrial revolution [1,2,3]. Consequently, the capture, storage, and transformation of CO_2_ have increasingly become the focus of worldwide attention [4,5,6]. Meanwhile, the correlative exhaustion of fossil resources also stimulated a thriving development of biodiesel fuel, which led, in turn, to a glycerol (GL) surplus all over the world [7]. As a result, the over-produced glycerol has been the major by-product during biomass utilization and has contributed to more than 67% of the global GL manufacturing and processing industry, thereby giving rise to a decreased market competitiveness and increased disposal difficulties [8,9]. Hence, consumption of glycerol through a green and economical process seems to be attractive and urgently needed. Among a series of attempts that have been being undertaken for bio-glycerol valorization, the pathway of coupling CO_2_ to synthesize glycerol carbonate (GC) proves to be attractive, which succeeds in realizing a value-added conversion of the two cheap and waste raw materials with high atom efficiency (87%) [10,11,12]. However, due to the thermodynamics limitation, this sustainable development strategy severely suffered from the apparent contradiction between glycerol conversion and GC yield [13].

A large number of homogeneous (Cs_2_CO_3_ [14], ^n^Bu_2_SnO [15], and KI [16]) and heterogeneous catalysts (La_2_O_2_CO_3_-ZnO [17], CeO_2_-ZrO_2_ [18], zeolite Y [19], and ZIF-8 [20]), have been reported and designed to improve the carboxylation of GL with CO_2_. Whereas by feat of harsh reaction conditions (high temperature and pressure) and indispensable dehydrating agents (mainly 2-cyanopyridine or acetonitrile), both GL conversion and GC yield are still far from satisfactory under the current catalytic reaction system. This can be owing to, on the one hand, the poor recyclability and separation difficulties of homogeneous catalysts, and on the other hand, the inadequate acid-base sites and specific surface area of heterogeneous catalysts. While for the latter, with competitive GC yields and notable advantages in easing product separation and purification, it has been proved to be worth more in-depth research. The future efforts can be directed towards enhancing the surface adsorption and activation of CO_2_ and glycerol of heterogeneous catalysts by enlarging the active specific surface area, creating high density of Lewis acid and basic sites.

Zeolites characterized by a high specific surface area and adjustable acid/basic sites have been considered as potential candidates for catalyzing glycerol direct carbonylation with CO_2_ [21,22,23]. Especially, titanosilicate zeolite ETS-10, that enjoys unique a framework and chemical compositions deserves the attention [24,25]. As one of the few zeolites that are with intrinsic Lewis basic sites, ETS-10 has proved to be promising in CO_2_ adsorption and activation, during which the coordinately unsaturated TiO_6_ building unit played the decisive role [26,27]. Furthermore, it has also displayed extraordinary performance for glycerol transesterification as a heterogeneous base catalyst, which was turned out to depend mainly on the interaction between Lewis basic sites (TiO_6_^2−^) and glycerol hydroxyl groups [28]. Added to these advantages is the porous structure that comprises chains of corner-sharing TiO_6_ octahedra surrounded by SiO_4_ tetrahedra, leading to an effective pore size of about 0.8 nm [29,30]. Thus, considering the actual molecular dimension of CO_2_ and glycerol is 0.12 nm and 0.62 nm [31], respectively, there will be easy access for them to catalytic active sites if ETS-10 is employed as catalyst.

Hence, in this work, a series of ETS-10 based catalysts were prepared and used for catalytic transformation of CO_2_ and GL, among which Co/ETS-10 exhibited the highest catalytic activity (GL conversion: 35.0%; GC yield: 12.7%). Notably, herein, the obtained result is even obviously superior to those that have been reported previously. This is largely due to the multiple active centers constructed by the strong interaction between the cobalt species and the unique ETS-10 framework, which indeed fulfills the requirement for simultaneous activation of CO_2_ and glycerol. Therefore, CO_2_ molecules can be adsorbed and activated by the Lewis basic sites on the ETS-10 zeolite surface, which were then inserted into the cobalt glycerate generated by the Co species with Lewis acidity to form the crucial intermediate species for the following intramolecular rearrangement process, and finally produce the target product GC.

## 2. Results and Discussion

### 2.1. Catalyst Characterization

The XRD patterns of various ETS-10 based catalysts are shown in Figure 1a, which exhibit well-resolved peaks related to the ETS structure between 5° and 45°. While comparing those metal introduced ETS-10 zeolite catalysts with the pristine one, the decreased crystallinity suggested that the incorporation of metal species did indeed threaten the integrity of the zeolite framework to varying degrees, which can be further confirmed by the N_2_ adsorption-desorption analysis data (Table 1). The BET-specific surface area reduced to 283.4 m^2^/g (Zn/ETS-10) from 324.4 m^2^/g (ETS-10) gave a good interpretation of the negative influence on the ETS-10 structure integrity due to the metal species and their different interactions with zeolite support. It is worth noting that even though the actual metal contents detected by ICP analysis were around 4%, there seems no metal particles can be observed in the XRD patterns, indicating the high dispersion of the metal species.

Figure 1b reveals the basic properties of different ETS-10 zeolite-based samples determined by CO_2_-TPD. Obviously, there were alkaline sites of different strength distributed on the catalyst, most of which were characterized by medium-strong alkalinity with the CO_2_ desorption temperature above 400 °C This can be related to the unidentate carbonates formed on O^2−^ ions that are located in the structural unit of the Ti-O-Ti chain [32]. Despite the relatively weak peak intensities, two broad peaks centered at about 80 °C and 180 °C were presented for all eight samples, which resulted from the interactions between CO_2_ and the weak basic surface hydroxyl groups on ETS-10 zeolite [26]. In addition, the quantitative analysis of CO_2_-TPD test was also performed to help visualize the overall CO_2_ adsorption capacity of different catalyst samples (Table 1). Firstly, according to the comparison between ETS-10 and those metal-based samples, the most visible changes were the distribution of the desorption temperature in the high region. For ETS-10, the most salient desorption peak was at 620 °C, while all the others showed an evident lower temperature distribution (<600 °C). Nevertheless, the increase of the total desorption quantity of CO_2_ was visible to the naked eye for the metal introduced samples. Ni/ETS-10 was no doubt the most prominent one with the highest amount of desorbed CO_2_ (1.41 mmol/g), indicating a promising CO_2_ adsorption capacity. Fe/ETS-10, Ce/ETS-10, and Cu/ETS-10 showed similar basic characteristics with the desorption temperature distribution and the total CO_2_ desorption quantity being parallel. Zn/ETS-10 and Co/ETS-10 were the two extraordinary samples with the quantitative result being not comparable. For Zn/ETS-10, it benefited from the strong interaction between zinc species and ETS-10 zeolite, which was mainly featured in the dominance of strong alkaline sites. However, meanwhile, the interaction was in fact a two-edged sword, which has proved to be adverse to keeping the catalyst structure integrity. As a consequence, the seriously decreased crystallinity (90.2%) brought disadvantages to the physico-chemical properties of the corresponding catalyst. When it came to Co/ETS-10, it unexpectedly exhibited the lowest CO_2_ desorption amount of 0.44 mmol/g among a series of metal-introduced ETS-10 zeolite catalyst samples with the desorption temperature shifting even below 500 °C. Apparently, the intriguing anomaly could not be simply attributed to the decreased catalyst crystallinity, which should be closely related to the metal species themselves and their distinct interaction with the zeolite support. In fact, although the CO_2_ adsorption capacity of Co/ETS-10 was inferior to other catalyst samples, the total amount of desorbed CO_2_ was almost 15 times more than that of Co_3_O_4_ reported previously (about 0.03 mmol/g) [33,34], further illustrating that the integration of the cobalt species and ETS-10 zeolite did generate a peculiar interaction. Given the relative specificity of Co/ETS-10, further investigations were carried out for taking a closer look at the interaction between metal Co species and ETS-10 zeolite.

Figure 2 shows the SEM images of as-synthesized ETS-10 and Co/ETS-10 samples. On the whole, both samples displayed structured crystals with a typical bipyramidal truncated shape, and the layer stacked ETS-10 crystals were all with uniform size (4–6.5 μm) and minor surface defects. This indicates that the incorporation of the Co species into ETS-10 does not cause any observable structure destructions to the zeolite support, which also demonstrates their moderate interaction and is in line with the XRD and N_2_ adsorption-desorption results.

The surface composition and chemical bonding states of the metal active sites on Co/ETS-10 were then determined by XPS. As can be seen from Figure 3, there are six characteristic peaks from the Co *2p* XPS spectrum. The peaks at 779.6 eV and 794.2 eV are affiliated to Co *2p*_3/2_ and Co *2p*_1/2_ for Co^3+^ species. Another two characteristic peaks appearing at 781.4 and 796.1 eV are attributed to Co *2p*_3/2_ and Co *2p*_1/2_ for Co^2+^ species, which are accompanied by two shake-up satellite peaks located at 786.0 eV and 802.9 eV. Interestingly, when compared with the XPS spectra of Co_3_O_4_ reported previously [35,36], the peak positions for Co^3+^ species in Co/ETS-10 are shifted toward lower binding energies, suggesting a possible existence of electron transfer between the Co species and the ETS-10 zeolite support. Furthermore, according to the semi-quantitative analysis, the surface concentrations of Co^2+^ for Co/ETS-10 is 73%, higher than that for Co_3_O_4_ reported (69%), indicating the lower average oxidation valence state of the cobalt species presented in Co/ETS-10 [37,38]. This may be related to the coordination interaction between Co species and TiO_6_ building units with two negative charges in ETS-10 zeolite.

### 2.2. Catalytic Activity

A series of metal-based ETS-10 zeolite catalysts were firstly used for the transformation of glycerol and CO_2_ to GC (Figure 4a). A blank control experiment was also carried out with virtually no GC formed, demonstrating the necessity of introducing a catalyst for promoting CO_2_ and glycerol activation-conversion. Moreover, it also substantiated the importance for employing the appropriate solvent. Herein, acetonitrile, one of the most often used solvents, proved to play a fundamental role in activating glycerol, leading to nearly 17% glycerol being converted. This can be owing to the acetamide generated from the hydrolysis of acetonitrile, that can induce the coupling reaction to remove the by-product H_2_O, which thus benefited by breaking the strict thermodynamic limitation and facilitated the reaction equilibrium shifting to right [39,40]. When compared to the experimental run with pure ETS-10 adopted, a marked improvement of both GL conversion (24.5%) and GC selectivity (9.8%) was obtained, which can be attributed to the enhanced CO_2_ adsorption-activation capacity that resulted from the unique framework of ETS-10 zeolite. Indeed, it has been reported in our previous work, that the peculiar structure unit of the -Ti-O-Ti- chain that is characterized by the extraordinary and strong donor capability, did meet the demands of CO_2_ activation, therefore, greatly promoted the correspondingly catalytic performance for CO_2_ transformation [27,41]. For all supported catalysts, most of them showed higher GL conversion and GC yield than the metal-free catalyst samples, except Ni/ETS-10, which may be due to its excessive basicity. Obviously, Co/ETS-10 and Zn/ETS-10 each occupied the highest GL conversion and GC selectivity, respectively. As a matter of fact, zinc-based catalysts have been widely reported and recognized as the most potential candidate for synthesizing GC from glycerol and CO_2_ [17,20,42]. While in this work, it seems that Zn/ETS-10 is slightly inferior to Co/ETS-10 with 0.5% less of the total GC yield. Considering the impressive GC selectivity, the disadvantages of Zn/ETS-10 in GL conversion may be related to the lower degree of glycerol and CO_2_ activation. This can be associated directly with the crystallinity deterioration of ETS-10 zeolite resulting from Zn introduction, which gave rise to the strong electronic interaction between zinc species and O atoms in the zeolite skeleton. Thus, a distortion of the Ti-O-Ti chain took place, which then resulted in the structure damage of ETS-10 zeolite and brought negative influence on the relevant catalytic performance. In other words, further investigations in developing more moderate and effective ways to introduce zinc species are need in the future. For Cu/ETS-10, Ce/ETS-10, and Fe-ETS-10 catalysts, they were all highly enthusiastic about converting glycerol, but their capacity in boosting GC selectivity was too common to be on par with Co/ETS-10 and other catalysts that have been reported [39,40,43]. Another notable sample was Zr/ETS-10, of which the introduced Zr species seems to play little or no role in the GC formation. This result can be interpreted as the composite demands for a certain amount of basic sites, the binding energy of the metal species and their interaction with catalyst supports. Though the incorporation of Zr^4+^ into ETS-10 zeolite endowed the catalyst with more medium and strong basic sites, the fairly low binding energy seriously pulled down the activation of glycerol [18,44,45,46].

The effect of pressures on the carbonylation of glycerol with CO_2_ was then investigated. As can be shown in Figure 4b, both the glycerol conversion and GC yield monotonically increased with the initial reaction pressure. This can be attributed to, on the one hand, as the main reaction feedstock, the presence of more CO_2_ can bring it into sufficient contact with other reactant molecules and catalytic centers, which thus is favorable for the reaction to proceed in the desired direction. On the other hand, higher pressure provided an easier dissolution of CO_2_ molecules in the acetonitrile solution, leading to convenient access for CO_2_ adsorption-activation. Consequently, with the pressure increased from 1.0 MPa to 5.0 MPa, the conversion of glycerol dramatically increased from 20.4% to 29.4%. For the GC yield, the growth was also obvious before the reaction pressure was increased to 4.0 MPa, after which it slowed down with only 0.8% more GC yield was obtained. Therefore, taking the actual requirements of industrial application, a relatively low operating pressure is safer and cost-effectiveness.

Figure 4c shows the effect of the reaction temperature on the carbonylation of glycerol with CO_2_. As a typical exothermic reaction, the glycerol conversion deservedly benefited from the elevated temperature, which can be put down to the facilitated activation of low-energy molecules and lowered reaction energy barrier with temperature increased. Nevertheless, the selectivity of GC decreased significantly if the reaction temperature increased higher than 170 °C. This is because, on the one hand, the glycerol dehydration reaction can be intensified at a high temperature to produce glycidyl glycerol. On the other hand, the produced acetamide from acetonitrile hydrolysis will be further hydrolyzed to acetic acid, which then reacts with glycerol and leads to the formation of by-products glyceryl monoacetate and glyceryl diacetate. Moreover, according to the previous literature reports, the formed GC could be easily decomposed or converted in the presence of amine groups. Consequently, an excessive increase of reaction temperature will go against the GC selectivity, and it seems to be optimal to set the reaction temperature at 170 °C in this work.

It is widely known that the amount of catalyst definitely works for improving the catalytic performance by adjusting the active sites. Typically, the more catalysts added, the more the active sites are accessible to reactants. As shown in Figure 4d, increasing the catalyst dosage from 0.1 g to 0.25 g, a growth leap in glycerol conversion and GC selectivity took shape, after which the rising tendency came to a standstill. This can be demonstrated as being available for more and more indispensable active sites that remarkably enhanced the activation of CO_2_ and glycerol and facilitated their subsequent combination to efficiently produce GC. In spite of being beneficial to promoting catalytic reactivity, there seems no need to blindly increase the catalyst’s quantity to more than 0.25 g, where the catalytic active sites in the reaction system are perfectly adequate.

As presented in Figure 4e, even though there was obvious growth that occurred on the glycerol conversion with the reaction time prolonged, the rise in GC selectivity unexpectedly derailed, which was maximized at 6 h and then began declining gradually accompanied by an increased formation of glycerol monoacetate. This may be related to the side reaction of the active hydroxyl group on GC and acetic acid from CH_3_CN hydrolysis. Additionally, it is possible that with the presence of more and more GC in the reaction system, self-polymerization took place to produce oligomers under the given reaction conditions, which accordingly brought down the GC yield.

Thanks to the crucial role that CH_3_CN played in positively affecting the reaction process, the influence of the CH_3_CN dosage was then explored for optimization. Firstly, it can be seen from Figure 4f that with no acetonitrile used as dehydrating agent, the glycerol was less likely to be converted into GC. As the amount of acetonitrile was increased from 3 mL to 5 mL, either the conversion of glycerol or the selectivity of GC increased significantly, after which the reactivity enhancement appeared to not come into play by keeping on increasing the usage of CH_3_CN. Admittedly, the transformation of CO_2_ and glycerol to GC can be improved by the enhanced hydrolysis efficiency from excess CH_3_CN, however, acetic acid would be produced at the same time and reacted with glycerol, leading to the generation of undesired by-products and the decline of GC selectivity. Therefore, a moderate amount of 5 mL CH_3_CN is exactly good for GC synthesis.

Further studies on the catalyst prepared by various active metal species mentioned above for the carbonylation reaction of glycerol with CO_2_ were then reimplemented under the optimized reaction conditions of 170 °C, 4 MPa, 5 mL CH_3_CN, 6 h, and 0.25 g catalyst (Table 2). Clearly, all catalysts enjoyed a significant improvement of glycerol conversion and GC yield, among which Co/ETS-10 was still comprehensively superior. To further obtain a better insight into the distinguished catalytic performance of Co/ETS-10, a comparison with Co_3_O_4_ was conducted, which has been reported with dominantly advantageous binding energy that greatly benefited glycerol activation [47]. As a consequence, Co_3_O_4_ showed a comparable glycerol conversion with Co/ETS-10. While the evident weakness in yielding GC of Co_3_O_4_ reconfirmed the significance of ETS-10 zeolite, which has proved to be indispensable for adsorption and activation of both glycerol and CO_2_. What is more, as can be seen from Table 3, the catalytic performance of Co/ETS-10 not only preceded those catalysts mentioned above, but also was preeminent among various catalysts reported previously [17,48,49,50].

On account of the discussion above and referring to those previous reports [20,40,51,52], a feasible mechanism was proposed for the coupling process of GC synthesis. As is shown in Figure 1, the oxygen atom on the hydroxyl group of glycerol attacked the Co species to produce cobalt glycerate and water molecules as main intermediates. At the same time, the activated CO_2_ adsorbed on the basic sites (TiO_6_^2−^) of ETS-10 zeolite was inserted into the cobalt glycerate to form another important intermediate of the seven-membered ring, which then underwent intramolecular rearrangement for the target product GC coming into being. This process proceeded smoothly mainly thanks to the multiple catalytically active centers constructed by the suitable interaction between the cobalt species and the ETS-10 framework, which indeed fulfilled the requirement for simultaneous activation of CO_2_ and glycerol with high efficiency.

### 2.3. Catalyst Stability

Catalyst stability is important for practical industrial applications. Hence, the stability of the Co/ETS-10 catalyst after eight cycles under the desired reaction conditions was evaluated. As revealed in Figure 5, Co/ETS-10 proved itself to be reasonably stable with less than 3% decline in glycerol conversion and glycerol carbonate yield. Notably, the glycerol conversion and glycerol carbonate yield can even be sustained at a consistent level during the first five test runs.

## 3. Experimental Methods

### 3.1. Materials

Cerium nitrate hexahydrate (AR, 99%) and zirconium nitrate pentahydrate (AR, 99%) were purchased from Macklin Biochemical Co., Ltd. (Shanghai, China). Iron nitrate nonahydrate (AR, ≥98.5%), cobalt nitrate hexahydrate (AR, ≥99%), nickel nitrate hexahydrate (AR, 98%), cupric nitrate trihydrate (AR, 99%), zinc nitrate hexahydrate (AR, 99%), glycerol (ACS, ≥99.5%), and tetraethylene glycol (AR, 99%) were supplied by Aladdin Chemistry Co., Ltd. (Shanghai, China). Acetonitrile (AR, 99%) and ethanol (AR, 99%) were obtained from Sinopharm Chemical Reagent Co., Ltd. (Beijing, China). Other reagents were of analytical grade and all of the reagents were used as received without further purification.

### 3.2. Preparation of ETS-10 Zeolite-Based Catalysts

ETS-10 was prepared by a hydrothermal approach reported previously with a molar composition of 1.0 TiO_2_/7.1 SiO_2_/4.4 Na_2_O/1.9 K_2_O/164 H_2_O [27]. Typically, the synthesis was started from the dropwise addition of 5.9 g TiCl_3_ solution (15–20 wt% in 30 wt% HCl) to the alkaline solution consisting of 9.3 mL waterglass solution (SiO_2_/Na_2_O molar ratio: 3.2, density: 1.2 g/cm^3^) and 10 mL aqueous NaOH (3.5 mol/L). Subsequently, 5.2 mL KF aqueous solution (5.8 mol/L) was introduced to obtain the final slurry gel, which was then transferred into the Teflon-lined autoclave and crystallized at 230 °C for 72 h. The products were recovered by filtration, washed with deionized water, dried at 100 °C and calcined in air at 450 °C for 5 h. The obtained ETS-10 zeolite samples were used to prepare series of metal-based catalysts by an incipient impregnation method with a 5 wt% metal loading. Taking Co/ETS-10 as an example, 1.0 g ETS-10 was mixed uniformly with 0.5 mL Co(NO_3_)_2_ aqueous solution (1.7 mol/L), after which the resulted solid product was air dried overnight. A further drying at 120 °C for 12 h was then carried out before being calcined at 450 °C for 5 h to afford the final Co/ETS-10 catalyst sample. By the same way, other catalysts were also prepared with nitrate precursors, including Ce(NO_3_)_3_·6H_2_O, Zr(NO_3_)_4_·5H_2_O, Fe(NO_3_)_3_·9H_2_O, Ni(NO_3_)_2_·6H_2_O, Cu(NO_3_)_2_·3H_2_O, and Zn(NO_3_)_2_·6H_2_O.

### 3.3. Catalyst Characterization

Phase identification of all samples was performed by powder X-ray diffraction (XRD) measurements using a Rigaku powder X-ray diffractometer (D/Max 2500, Rigaku, Tokyo, Japan). The diffraction peaks were scanned from 5° to 45°. The actual metal contents were determined by inductively coupled plasma optical emission spectrometry (ICP-OES-PerkinElmer Optima 3300DV). The specific surface area of the catalysts was measured at −196 °C using an adsorption apparatus (Micromeritics ASAP 2460, Micromeritics, Norcross, GA, USA) and was estimated from the adsorption data using the Brunauer–Emmett–Teller (BET) equation. The morphology was observed by field-emission scanning electron microscopy (Hitachi S-4800, Hitachi, Tokyo, Japan).

The catalyst’s basicity was evaluated using the temperature-programmed desorption of CO_2_ on a Micromeritics ASP 2920 instrument (Micromeritics, Norcross, GA, USA), by which the corresponding CO_2_ adsorption capacity was also calculated. A 200 mg sample was placed in a quartz tube and then the temperature was raised from room temperature to 450 °C for 2 h, after which it cooled to 100 °C to allow CO_2_ gas to be passed through for 30 min. Following this was the physical removal of CO_2_ by flowing helium for 2 h at 100 °C with the overall flow rate of the gas being fixed at 10 m^3^/min. Subsequently, the spectra were recorded at a heating rate of 10 °C/min from 100 °C to 650 °C. The oxidation states of the metal atoms were determined by X-ray photoelectron spectroscopy (XPS), which was performed on a VG ESCA Lab 250 photoelectron spectrometer (Thermo Fisher Scientific Company, Waltham, MA, USA) using an Al Kα X-ray resource (hν = 1486.6 eV).

### 3.4. Catalytic Activity Test

A typical implementation with acetonitrile being used as a dehydrating agent is as follows. Initially, 4.6 g glycerol, 5 mL acetonitrile, and 0.25 g catalyst were mixed in a high-pressure stainless-steel reactor (200 mL). Then, 1 mL tetraethylene glycol was added into the reaction mixture as an internal standard substance. Then, the reactor was sealed and purged with CO_2_ 3 times before being pressurized to 4 MPa. Finally, the autoclave was heated to the reaction temperature and maintained for a certain reaction time with stirring (600 r/min). Following the reaction, the reactor was firstly cooled to room temperature and depressurized. The liquid fraction separated by centrifugation was quantificationally and qualitatively analyzed by gas chromatograph (GC) and GC-mass instruments equipped with FID detectors. The remaining solid catalyst was washed with ethanol, dried, and calcinated for recycling.

For product’s analysis, the GC oven temperature was set at 120 °C and increased by a ramping rate of 25 °C/min until it reached 230 °C. The FID and injection temperatures were set at 300 °C. The reactant conversion (C_GL_), product yield (Y_GC_), and selectivity (S_GC_) were calculated using the following formulas:$$\mathrm{Conversion}\;\left(\%\right)=\;\frac{\mathrm{initial}\;\mathrm{mole}\;\mathrm{of}\;\mathrm{glycerol}\;-\mathrm{mole}\;\mathrm{of}\;\mathrm{glycerol}\;\mathrm{unconverted}}{\mathrm{initial}\;\mathrm{mole}\;\mathrm{of}\;\mathrm{glycerol}}\times 100\%$$
$$\mathrm{Yield}\;\left(\%\right)=\frac{\mathrm{mole}\;\mathrm{of}\;\mathrm{glycerol}\;\mathrm{carbonate}}{\mathrm{theoretical}\;\mathrm{mole}\;\mathrm{of}\;\mathrm{glycerol}\;\mathrm{carbonate}}\times 100\%$$
$$\mathrm{Selectivity}\;\left(\%\right)=\;\frac{\mathrm{mole}\;\mathrm{of}\;\mathrm{glycerol}\;\mathrm{carbonate}\;\mathrm{produced}}{\mathrm{initial}\;\mathrm{mole}\;\mathrm{of}\;\mathrm{glycerol}\;-\mathrm{mole}\;\mathrm{of}\;\mathrm{glycerol}\;\mathrm{unconverted}}\times 100\%$$

## 4. Conclusions

A series of ETS-10 zeolite-based catalysts were used to investigate the coupling process of CO_2_ and glycerol to produce glycerol carbonate. For comparison, different metal species were employed and studied, among which Co/ETS-10 catalyst presented obvious advantages in both glycerol conversion and glycerol carbonate selectivity. Despite the excellent GC selectivity of Zn/ETS-10, the destructive effect of Zn on the structure of ETS-10 is not negligible, leading to the impaired competitiveness. This deserves further investigation in the future. For Cu/ETS-10, Ce/ETS-10, and Fe/ETS-10 catalysts, the obtained GC yield was mediocre. The excessive basicity of Ni/ETS-10 and the low binding energy of Zr^4+^ in Zr/ETS-10 were responsible for their own poor catalytic performance. Comprehensive analysis revealed that the presence of moderate Lewis basic sites on ETS-10 zeolite and their synergistic effect with the introduced Co species were vital for the activation of CO_2_ and glycerol to form the desired GC. As a result, Co/ETS-10 with fair structural integrity, appropriate Lewis basic, and acid strength was proved to be the highest reactive among the studied catalysts for GC formation, giving a 35.0% glycerol conversion and 12.7% GC under a mild condition of 170 °C, 4.0 MPa initial pressure, 5 mL CH_3_CN, and 6 h. The last but not least, the catalytic performance of Co/ETS-10 can be well maintained at 170 °C after eight cycles with only less than 3% decline in glycerol conversion and GC yield, indicating its excellent stability and reusability.

## Data Availability

The data that support the findings of this study are available from the corresponding author upon reasonable request.

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
