# Peer review of "Transformation of CO2 with Glycerol to Glycerol Carbonate over ETS-10 Zeolite-Based Catalyst"

_molecules, 2023, doi:10.3390/molecules28052272_

Round 1
Reviewer 1 Report
The manuscript has been reviewed and it needs substantial revision before further processing.
1. Mention the metals which are used during the study in the experimental.
2. Give the method for preparation of metal based ETS-10 catalyst in detail.
3. Give the characterization details of all synthesized metal catalysts (SEM, EDX)
4. The authors should explain other metal HTS-10 catalyst results in conclusion and abstract.
Author Response
The manuscript has been reviewed and it needs substantial revision before further processing.
- Mention the metals which are used during the study in the experimental.
Thank you very much. The detailed introduction of metals used during our experiments has been added in the section "Preparation of ETS-10 zeolite-based catalysts".
- Give the method for preparation of metal based ETS-10 catalyst in detail.
Thanks very much. To be more clearly and in accordance with the reviewer concerns, the preparation process of metal-based ETS-10 zeolite catalyst has been describe in detail.
- Give the characterization details of all synthesized metal catalysts (SEM, EDX).
Thank you for your valuable advice. The details for all the characterizations that have been performed was given in this paper. Considering that the actual metal contents has been detected by ICP, there is no strong necessity to conduct SEM-EDX for those unreduced metal species with no significant morphology change.
- The authors should explain other metal HTS-10 catalyst results in conclusion and abstract.
Thanks very much. The explanation of the results for other metal-based ETS-10 zeolite catalysts has been added in the section of Abstract and Conclusions and marked in red.
Reviewer 2 Report
In this manuscript, the authors demonstrated a method of transformation of CO2 with glycerol-to-glycerol carbonate over ETS-10 zeolite-based catalyst. The catalytic GL conversion at 170 degrees Celsius could reach 35.0 % and a 12.7 % yield of GC was obtained on Co/ETS-10 with CH3CN as a dehydrating agent. Analysis revealed that the presence of moderate basic sites for CO2 adsorption-activation played an important role in regulating catalytic activity. Besides, the appropriate interaction between cobalt species and ETS-10 zeolite was also significant for improving the glycerol activation capacity. A mechanism was proposed. Moreover, the recyclability of Co/ETS-10 was also measured.
Although the current method is not that novel, and the performance is not great to some extent, considering the detailed studies the authors provided, I support its publication in this journal.
One minor issue is that the authors might need to improve their academic writing a little bit.
Author Response
In this manuscript, the authors demonstrated a method of transformation of CO2 with glycerol-to-glycerol carbonate over ETS-10 zeolite-based catalyst. The catalytic GL conversion at 170 degrees Celsius could reach 35.0 % and a 12.7 % yield of GC was obtained on Co/ETS-10 with CH3CN as a dehydrating agent. Analysis revealed that the presence of moderate basic sites for CO2 adsorption-activation played an important role in regulating catalytic activity. Besides, the appropriate interaction between cobalt species and ETS-10 zeolite was also significant for improving the glycerol activation capacity. A mechanism was proposed. Moreover, the recyclability of Co/ETS-10 was also measured. Although the current method is not that novel, and the performance is not great to some extent, considering the detailed studies the authors provided, I support its publication in this journal. One minor issue is that the authors might need to improve their academic writing a little bit.
Thank you very much. The inappropriate expressions in this work have been corrected and marked in red for a scientific communication.
Round 2
Reviewer 1 Report
accept